# Metabolome and Transcriptome Profiling Unveil the Mechanisms of Polyphenol Synthesis in the Developing Endopleura of Walnut (*Juglans regia* L.)

**DOI:** 10.3390/ijms23126623

**Published:** 2022-06-14

**Authors:** Ruimin Huang, Ye Zhou, Feng Jin, Junpei Zhang, Feiyang Ji, Yongchao Bai, Dong Pei

**Affiliations:** State Key Laboratory of Tree Genetics and Breeding, Key Laboratory of Tree Breeding and Cultivation of the State Forestry and Grassland Administration, Research Institute of Forestry, Chinese Academy of Forestry, Beijing 100091, China; huangruimin1@gmail.com (R.H.); zhou.ye@caf.ac.cn (Y.Z.); fjin3523@gmail.com (F.J.); zhangjunpei@caf.ac.cn (J.Z.); jifeiyangcaf@gmail.com (F.J.); baiychao@bjfu.edu.cn (Y.B.)

**Keywords:** pellicle, endopleura, flavonoid, tannins, MYB

## Abstract

Walnut (*Juglans regia* L.) is an important woody nut tree species, and its endopleura (the inner coating of a seed) is rich in many polyphenols. Thus far, the pathways and essential genes involved in polyphenol biosynthesis in developing walnut endopleura remain largely unclear. We compared metabolite differences between endopleura and embryo in mature walnuts, and analyzed the changes of metabolites in endopleura at 35, 63, 91, 119, and 147 days after pollination (DAP). A total of 760 metabolites were detected in the metabolome, and the polyphenol contents in endopleura were higher than those in embryos. A total of 15 types of procyanidins, 10 types of kaempferol glycosides, and 21 types of quercetin glycosides that accumulated during endopleura development were identified. The analysis of the phenylpropane metabolic pathway showed that phenylalanine was gradually transformed into proanthocyanidins and other secondary metabolites with the development of endopleura. A total of 49 unigenes related to polyphenol synthesis were identified by transcriptome analysis of endopleura. The expression patterns of *PAL*, *C4H*, *4CL*, *CHS*, *CHI*, *F3H*, *LDOX*, and *ANR* were similar, and their expression levels were highest in endopleura at maturity. Transcriptome and metabolome analysis showed that endopleura rapidly synthesized and accumulated polyphenols during maturation. Moreover, the transcription factor MYB111 played an important role in synthesizing polyphenols in endopleura, and its expression pattern was positively correlated with the accumulation pattern of quercetin, kaempferol, and proanthocyanidins. *MYB111* was co-expressed with *NAP*, *NAC*, *ATR1*, and other genes related to cell senescence and abiotic stress response. Our study analyzed the composition and molecular synthesis mechanism of polyphenols in walnut endopleura, and provided new perspectives and insights regarding the nutritional research of walnut nuts.

## 1. Introduction

Walnut (*Juglans regia* L.) belongs to the Juglans family and is an ancient and widely planted woody oil tree species [1]. Walnut kernels are both delicious and nutritious. Each 100 g walnut kernel contains 60–70 g of oil, 15–20 g of protein, 10 g of carbohydrates, and many biologically active nutrients, such as phenolic acid and riboflavin [2]. Notably, walnuts are particularly rich in polyphenols, which are higher found in higher levels than other nuts (i.e., almonds, hazelnuts, pistachios and peanuts) [3]. Moreover, walnuts have a higher antioxidant capacity and ellagic acid content than these nuts.

The polyphenols in walnut kernels have an important effect on quality. Some polyphenols can bind to the proteins in human saliva, giving and astringent taste [4], and the oxidation and degradation of polyphenols will change the color of walnut kernels. In addition, polyphenols have a strong anti-free radical ability, which can enhance immunity, provide resistance against atherosclerosis, and protect eyesight [5]. Recent studies have shown that walnut phenol extract can inhibit the self-renewal and proliferation of colon-cancer stem cells, enhance their differentiation ability, and have an anti-cancer effect [6]. An in vitro model of lipid peroxidation also proved that polyphenol-rich walnut extract can inhibit oxidative stress in type 2 diabetic mice [7].

Polyphenol synthesis in plants depends on the correct spatial and temporal activity of many gene products. The phenylpropane pathway is a general pathway for synthesizing polyphenols, and all metabolites containing the phenylpropane skeleton are directly or indirectly synthesized by this pathway. Phenylalanine ammonia-lyase (PAL) catalyzes phenylalanine to cinnamic acid, which is the enzyme in the first step of phenylpropane metabolism and the rate-limiting enzyme [8]. Subsequently, cinnamate 4-hydroxylase (C4H) catalyzes cinnamic acid to coumaric acid, and 4-coumarate-CoA ligase (4CL) catalyzes coumaric acid to 4-coumaroyl-CoA. Then, chalcone synthase (CHS) catalyzes one molecule 4-coumaroyl-CoA and three molecule malonyl-CoA to chalcone, and chalcone isomerase (CHI) catalyzes chalcone to naringenin. CHS and CHI are rate-limiting enzymes for flavonoid biosynthesis [9,10].

Flavanone-3-hydroxylase (F3H) catalyzes naringenin to dihydrokaempferol, and flavonoid 3′-hydroxylase (F3′H) catalyzes dihydrokaempferol to dihydroquercetin [11]. Subsequently, flavonol synthase (FLS) catalyzes dihydrokaempferol to kaempferol, and dihydroquercetin to quercetin. Dihydroflavonol 4-reductase (dihydroflavonol 4-reductase, DFR) can reduce dihydroquercetin to leucoanthocyanidin [12,13]. Then, leucoanthocyanidin dioxygenase (LAR) catalyzes leucoanthocyanidin to epicatechin, and leucoanthocyanidin dioxygenase (LDOX) and anthocyanin reductase (ANR) catalyze leucoanthocyanidin to epicatechin. Different amounts of catechin or epicatechin polymerize to form proanthocyanidins. Some studies have found that polyphenol accumulation can be increased by overexpression of *CHS*, *CHI*, *F3H*, *F3’H*, and *FLS* genes in many plants [14,15].

Some transcription factors regulate the biosynthesis of plant polyphenols [16]. The three types of transcription factors, MYB, bHLH, and WD40, can form MYB-bHLH-WD40 (MBW) protein complex, which binds to the promoters of the structural genes of the secondary metabolic synthesis pathway and regulates the accumulation of polyphenols [17]. In *Arabidopsis*, the expression of FLS1 is controlled by at least three R2R3-MYBs, including MYB12, MYB11, and MYB111 [18]. The transcription factors MYB11, MYB12, and MYB111 are similar in structure and belonged to subgroup 7 of the *Arabidopsis* R2R3-MYB gene family, and seedlings of triple mutants *myb11*, *myb12*, and *myb111* do not form flavonoids [18]. Moreover, MYB12 mainly controls the synthesis of flavonoids in the roots, while MYB111 mainly controls the synthesis of flavonoids in the cotyledons.

Plant secondary metabolites have important development value. Understanding the types and quantities of metabolites in plant tissues is the basis of their research and utilization. A previous study [19] used liquid chromatography coupled with electrospray ionization hybrid linear trap quadrupole-Orbitrap mass spectrometry (LC–LTQ-Orbitrap) to identify phenolic compounds in walnuts on a large scale. A total of 120 metabolites were initially identified, including hydrolyzed and condensed tannins, flavonoids, and phenolic acids. Wang et al. [20] reported that many phenolic metabolites accumulate in the endopleura of mature fruits, while lipids and amino acids and their derivatives mainly accumulate in the embryo. However, the reasons for high polyphenol content in walnut endopleura have not been investigated. In this study, we conducted metabolome and transcriptome profiling to identify the key genes related to polyphenol accumulation during endopleura development. NAP, NAC, ATR1, and other proteins related to cell senescence and abiotic stress response may promote the accumulation of polyphenols in walnut endopleura. Our study provided basic data for the research of the metabolite species and the formation mechanism of walnut endopleura.

## 2. Results and Discussion

### 2.1. Morphological Characteristics and Polyphenol Content

As shown in Figure 1A, walnut endopleura are developed from integument. Endopleura are attached to the ovary wall in the early stage of nut development (P1), and when the embryo develops, endopleura wrap the embryo (P2–P5). The endopleura of walnut in our study was yellow, and the embryo was white (B5). As shown in Figure 1B, the polyphenol content in endopleura increased from 5.68 mg GAE g^−1^ (P1) to 7.38 mg GAE g^−1^ (P2) as endopleura developed. During stage P2–P4, the polyphenol content changed little and was about 7.74 mg GAE g^−1^. In the mature walnut nut, endopleura had the highest polyphenol content (9.78 mg GAE g^−1^). The polyphenol content in the embryo was 2.22 mg GAE g^−1^ (B5), which was less than that in endopleura.

### 2.2. Comparison of DAMs between Endopleura and Embryo in the Mature Walnut Nut

In order to understand the metabolite differences between endopleura and embryo, we compared the endopleura and embryo of mature walnuts. There were 455 up-DAMs and 92 down-DAMs in endopleura (Figure 2A). Moreover, the up-DAMs in endopleura were mainly flavonoids, tannins, and phenolic acids, and the up-DAMs in embryos were mainly amino acids and derivatives (Figure 2B). It is noteworthy that 212 metabolites (i.e., platycaryanin A, quercetin-3-O-xyloside, and granatin A) were only found in the endopleura, but not in the embryo (Appendix A). Figure 2C shows twenty metabolites with the greatest difference in content between endopleura and embryo.

DAMs between endopleura and embryos were significantly enriched in three KEGG pathways (*p*-value < 0.05), which were flavonoid biosynthesis, flavonoid and flavonol biosynthesis, and arginine and proline metabolism (Table 1). Among them, flavonoid biosynthesis had the highest significance (Appendix A), including 28 DAMs, of which 27 metabolites accumulated in greater quantities in the endopleura than that in the embryo. In addition, 15 DAMs belonged to flavonoid and flavonol biosynthesis (Appendix A), of which 13 metabolites accumulated in greater quantities in the endopleura than that in the embryo. This result was similar to the study of Wang et al. [20].

We identified 330 polyphenols, including 132 flavonoids, 73 tannins, 111 phenolic acids, 5 lignans, and 9 coumarins (Appendix A). The mature walnut endopleura contained all 73 tannins detected in all samples, while only 37 tannins were found in embryos (Appendix A). Therefore, removing the endopleura can significantly reduce the astringency of the walnut kernel, but it also loses the nutrients in the endopleura. It is worth noting that the concentration of glansreginin A in the embryo is about seven times higher than that in the endopleura. Glansreginin A may be a walnut-specific metabolite, but whether it is present in pecan is debatable [21]. It has been reported that feeding glansreginin A to lipopolysaccharide (LPS)-induced inflammatory model mice can prevent LPS-induced abnormal hippocampal behavior, and that glansreginin A has neuroprotective effect brain [22].

### 2.3. Trend Analysis of DAMs and DEGs in Endopleura

We focused on the metabolite changes in the endopleura, and a total of 760 metabolites were identified in samples P1–P5. Principal component analysis (PCA) of fifteen endopleura samples was performed and it was found that the two principal components explained 73.4% of the total variance. The three samples were, at the same stage, close to each other, indicating that there was a high consistency between the three biological replicates (Figure 3A). A total of 632 DAMs were identified by pairwise comparison between samples at each stage (Appendix A). The cluster analysis based on DAMs is shown in Figure 3B. The results showed that the 15 samples could be divided into two groups: P1 and P2 constituted the first group, and P3, P4, and P5 formed the second group.

PCA of the gene expression profiles showed that the two principal components explained 37.6% of the total variance (Figure 3A). A total of 6300 DEGs were identified by pair comparison between samples at each stage (Appendix A). Cluster analysis is shown in Figure 3B, and the results showed that more DEGs were highly expressed in P1 sample than that in other samples. The 15 samples could be divided into three groups: P1 formed the first group; P2, P3, and P4 constituted the second group; and P5 formed the third group. Overall, the PCA of metabolome and RNA-seq was similar, but the clustering of DAMs and DEGs was not consistent. This may have been because the translation of RNA into protein then catalyzes the synthesis of metabolites, and RNA and metabolites have different accumulation patterns [23,24].

### 2.4. KEGG Enrichment Analysis of DAMs in Endopleura

According to the metabolite accumulation pattern in endopleura, the 632 DAMs could be divided into nine subclasses (Figure 4A). Subclass 4 was the largest subclass, containing 111 metabolites. The metabolite content was highest in P1 sample, and the KEGG enrichment pathway was linolenic acid metabolism. Subclass 3 was the second largest subclass, containing 110 metabolites. The metabolite content was highest in P5 sample, and the KEGG enrichment pathway was flavonoid biosynthesis and the phosphotransferase system (PTS). In addition, Subclass 1 contains 17 metabolites, and KEGG was also associated with flavonoid biosynthesis.

As shown in Figure 4B, 632 DAMs were detected in 12 categories, including flavonoids, tannins, phenolic acids, organic acids, alkaloids, quinones, terpenoids, nucleotides and derivatives, lignans and coumarins, lipids, and amino acids and derivatives. A total of 632 DAMs contain 114 flavonoids, 58 tannins, and 97 phenolic acids, among which 55 flavonoids, 28 tannins, and 20 phenolic acids belonged to Subclass 3. These results indicated that most polyphenol-related metabolites accumulated during endopleura development.

### 2.5. Metabolites Related to Polyphenol Synthesis in Endopleura

We identified 53 metabolites related to polyphenol synthesis, including 15 procyanidins, 10 kaempferol glycosides, and 21 quercetin glycosides (Table 2). Among these 53 metabolites, 20 belonged to Subclass 3, 13 belonged to Subclass 6, and 9 belonged to other subclasses. Some metabolites with high peak area (i.e., catechin, quercetin-3-O-α-L-arabinofuranoside, quercetin-3-O-rhamnoside, and quercetin-3-O-xyloside), may have had a high content in endopleura. The phenylalanine content decreased rapidly during P1–P3, while the contents of procyanidins, kaempferol glycosides, and quercetin glycosides increased rapidly during P2–P5. Correlation analysis of polyphenol-related metabolites in endopleura was conducted (Appendix A). The results showed that naringin chalcone was positively correlated with naringin, kaempferol, and quercetin. Catechin, epicatechin, proanthocyanidin, kaempferol, and quercetin were positively correlated with each other. Phenylalanine was significantly negatively correlated with downstream metabolites (i.e., naringin chalcone, catechin, epicatechin, proanthocyanidin, kaempferol, and quercetin), indicating that phenylalanine was gradually transformed into proanthocyanidins and other secondary metabolites during endopleura development.

### 2.6. Trend Analysis of DEGs in Endopleura

In order to further understand the mechanism of polyphenol synthesis in walnut endopleura, we focused on the expression trend of DEGs. The 6300 DEGs were divided into five modules using WGCNA (Figure 5A), and the number of DEGs and KEGG pathways of each module were listed in each module. The largest module (turquoise) contained 3394 DEGs whose expression was highest in the P1 sample, which included genes related to amino acid biosynthesis, carbon metabolism, and glycolysis/gluconeogenesis. The second largest module (blue) contained 1361 DEGs whose expression was highest in the P5 sample, which included genes related to phenylpropane biosynthesis, flavonoid biosynthesis, and galactose metabolism.

In order to understand the relationship between the gene expression patterns and metabolites, association analysis was performed using WGCNA (Figure 5B). Polyphenol content and polyphenol-related metabolites were highly correlated with blue and brown modules. In combination with KEGG pathway and module-trait correlation analysis, blue and brown modules were associated with polyphenol biosynthesis.

### 2.7. Identification of Genes Related to Polyphenol Synthesis

As shown in Table 3, a total of 49 unigenes related to polyphenol synthesis were identified, of which 15 belonged to the blue module, 5 belonged to the turquoise module, and 1 belonged to the yellow module. Most of the genes related to polyphenol synthesis were highly expressed in the P5 sample. Some genes related to polyphenol synthesis, such as *PAL*, *C4H*, *CHS*, *F3H*, *F3′H*, *LDOX,* and *ANR*, had FPKM values > 100. These highly expressed genes may play an important role in polyphenol synthesis in walnut endopleura.

In addition, some genes (i.e., *F3H*, *F3′H*, *DFR*, *LDOX*, *ANR,* and *LAR*) were expressed in only one copy, and these unigenes play an irreplaceable role in polyphenol synthesis in walnut endopleura. In the P5 sample, the expression level of polyphenol-synthesis-related genes (i.e., *PAL*, *C4H*, *4CL*, *CHS*, *CHI*, *F3H*, *F3′H*, *DFR*, *LDOX*, *ANS*, *LAR*, and *ANR*) was the highest, and the accumulation of polyphenol metabolites was the highest (Figure 6). The metabolomic and transcriptomic results showed that the mature stage was the key stage to accumulate polyphenol in five endopleura samples.

### 2.8. Transcription Factors Related to Polyphenol Synthesis

Transcription factors play an important regulatory role in tissue development and accumulation of metabolites. Among the 6300 DEGs, 529 genes belonged to transcription factors (Appendix A). There were 208, 148, 31, 52, and 90 transcription factors in turquoise, green, yellow, brown, and blue modules, respectively. Among them, 21 bHLH, 10 bZIP, 19 C2H2, 14 HD-ZIP, and 17 MYB family transcription factors belonged to the turquoise module, and they may be related to the early development of endopleura. Moreover, 20 ERF and 11 NAC family transcription factors belonged to the blue module, and they may be related to the late development of endopleura.

Some transcription factors, such as MYB111, TTG1, TT8, TT2, MYB5, EGL3, TT1, TT16, and TTG2, play key roles in plant polyphenol synthesis (Figure 7). As shown in Table 4, seven polyphenol-related transcription factors belonged to DEGs. *TT2* (Jr03G11832, Jr03G11834) and *TT16* (Jr05G10903) belonged to the turquoise module, *MYB5* (Jr14G11044) and *TT1* (Jr08G11835) belonged to blue module, *MYB111* (Jr02G12175) belonged to the brown module, and *MYB111* (Jr08G10063) belonged to the green module. *MYB5* (Jr14G11044) and *TT1* (Jr08G11835) had similar expression patterns with *PAL*, *C4H*, *4CL*, *CHS*, *CHI*, *F3H*, *LDOX,* and *ANR*, and the expression pattern of *MYB111* was positively correlated with the accumulation pattern of quercetin, kaempferol, and proanthocyanidins. Therefore, MYB111 (Jr02G12175), MYB5 (Jr14G11044), and TT1 (Jr08G11835) may play an important role in the polyphenol synthesis in walnut endopleura.

Previous studies have shown that MYB5 influences flavonoid composition in pomegranate [25], and overexpression of *MtMYB5* strongly induces proanthocyanidin accumulation in hairy roots of *Medicago truncatula* [26]. TT1 interacts with R2R3-MYB factors and affects early and late steps of flavonoid biosynthesis in the endothelium of *Arabidopsis thaliana* seeds [27]. MYB111 is a transcription factor of the R2R3-MYB subfamily, can bind to specific cis-elements in promoter of *CHS*, *F3H*, and *FLS1*, and, in turn, activates their transcription. Moreover, MYB111 is a regulator of flavonoid synthesis in *Arabidopsis*, and flavonoid accumulation was lower in mutants than in wild type plants, but higher in MYB111-overexpressed plants [28].

### 2.9. MYB111 Co-Expression Network Analysis

We found that the expression pattern of *MYB111* was positively correlated with the accumulation pattern of quercetin glycosides, kaempferol glycosides, and procyanidin in walnut endopleura. In order to reveal the regulation mechanism of MYB111, we constructed a co-expression network using WGCNA. As shown in Figure 8, the genes were ranked according to their weighted correlation with *MYB111* (Jr02G12175), and the top 37 genes were selected from the blue module. Among these co-expressed genes, NAP (Jr06G10616) and NAC6 (Jr03G10992) are members of the NAC transcription factor family, which can positively regulate chlorophyll degradation and cell death caused by leaf senescence [29]. ATR1 (Jr07G11827) belongs to the ERF transcription factor, which encodes the cyp450 reductase involved in the metabolism of phenylpropane [30].

Besides, PMAT2 (Jr03G10800) belongs to the HXXXD-type acyltransferase family, which encodes malonyltransferase that may play a role in phenolic xenobiotic detoxification [31]. FBX92 (Jr03G12761) protein has an F-box structure, which can reduce leaf size and cell proliferation rate [32]. NARS1 (Jr10G10527), together with NAC018/NARS2, regulates embryogenesis by regulating the development and degeneration of ovule integuments [33]. CYSB (Jr14G10243) encodes a protein with cysteine protease inhibitor activity, which can increase the tolerance to abiotic stresses (i.e., salt, osmotic, and cold stress) [34]. AEP3 (Jr15G11892) encodes the vacuolar processing enzyme belonging to a novel group of cysteine proteinases, which is up-regulated in association with various types of cell death and under stressed conditions [35]. In summary, these genes are mainly associated with stress and cell death.

### 2.10. qPCR Analysis of Polyphenol-Related Genes

Twelve key genes related to polyphenol synthesis were analyzed by qPCR. As shown in Figure 9, Panels A–I show the expression profiles of nine key genes in the phenylpropane metabolic pathway, and Panels J–L show the expression profiles of three transcription factors related to polyphenol synthesis. Except for the *TTG1* (Jr06G12013) gene that inhibits anthocyanin synthesis, these genes related to polyphenol synthesis were all highly expressed in the mature stage of endopleura (P5). Panel M shows that RNA-seq is highly correlated with qPCR data (R^2^ = 0.80, *p* < 0.01), indicating that the expression data obtained by RNA-seq is reliable.

Comparing the transcriptome data of endopleura with embryo [36], the expression levels of these genes in endopleura were much higher than those in embryos. These results support that the tissue-specific expression of these polyphenol-related genes resulted in the higher polyphenol content in endopleura than in embryos. The unigenes, *PAL* (Jr02G11537), *C4H* (Jr14G11389), *CHS* (Jr01G10656 and Jr02G10304), *F3H* (Jr07G12902), *F3′H* (Jr11G12560), *LDOX* (Jr11G11451), and *ANR* (Jr09G12363), were all expressed at the highest level in the mature endopleura. The high expression of structural genes was the reason why walnut endopleura was rich in polyphenols.

## 3. Materials and Methods

### 3.1. Plant Material

Walnut fruits were collected from the 25-year-old cultivar ‘LinZaoxiang’ walnut tree in Beijing, China (116°14′ E, 40°0′ N). The tree was pollinated on 15 April (0 DAP). Then, fruits were collected at 35 DAP and then every 28 days until 147 DAP (fruit maturity). Samples were collected at five stages from 21 May to 10 September 2019. At each stage of development, seven fruits were mixed into a biological repetition, with three repetitions per each stage. The sampling locations of endopleura and embryo are shown in Figure 1A. Then, samples were quickly frozen with liquid nitrogen and stored in a refrigerator at −80 °C.

### 3.2. Determination of Polyphenol Content

The samples were dried by freeze-dryer (Scientz-100F, Ningbo, China), and then ground (30 Hz, 1.5 min) to powder using a grinder (MM 400, Retsch, German). The polyphenol content was determined by the Folin–Ciocalteu method [37]. The 0.2 g sample was extracted in 10 mL 70% (*v*/*v*) methanol (Merck, Darmstadt, Germany) solution at 4 °C. The 0.1 mL extract was fixed to 0.5 mL with distilled water, and then Folin–Ciocalteu reagent (0.25 mL) and 20% sodium carbonate solution (1.25 mL) were added. After 40 min, the absorbance of the reaction solution was measured at 725 nm (DU730UV VIS, Beckman Coulter, CA, USA). The calibration curve (Appendix A) was prepared with gallic acid (Macklin, Shanghai, China), and mg gallic acid equivalent (GAE) g^−1^ FW was used to represent the polyphenol content. Each sample was performed in three biological replicates.

### 3.3. Metabonomic Analysis

The widely targeted metabolome was conducted by Metware Biotechnology Co., Ltd. (Wuhan, China). First, 100 mg of powder was weighed and extracted overnight with 1.0 mL 70% methanol solution at 4 °C. The data acquisition instrument system mainly included ultra-performance liquid chromatography (UPLC) (Shim-pack UFLC SHIMADZU CBM30A, Shimadzu, Kyoto, Japan) and tandem mass spectrometry (MS/MS) (Applied Biosystems 4500 QTRAP, AB Sciex, Boston, MA, USA). The analytical conditions were as follows: UPLC column, Agilent SB-C18 (1.8 µm, 2.1 mm * 100 mm); the mobile phase consisted of solvent A, pure water with 0.1% formic acid (Sigma, St. Louis, MO, USA), and solvent B, acetonitrile (Merck, Darmstadt, Germany) with 0.1% formic acid. Sample measurements were performed with a gradient program that employed the starting conditions of 95% A, 5% B. Within 9 min, a linear gradient to 5% A, 95% B was programmed, and a composition of 5% A, 95% B was kept for 1 min. Subsequently, a composition of 95% A, 5.0% B was adjusted within 1.1 min and kept for 2.9 min. The flow velocity was set as 0.35 mL per minute, the column oven was set to 40 °C, and the injection volume was 4 μL. The effluent was alternatively connected to an ESI-triple quadrupole-linear ion trap (QTRAP)-MS.

Linear ion trap (LIT) and triple quadrupole (QQQ) scans were acquired on a triple quadrupole–linear ion trap mass spectrometer (Q TRAP), AB4500 Q TRAP UPLC/MS/MS system, equipped with an ESI turbo ion-spray interface, operating in positive and negative ion mode and controlled by Analyst 1.6.3 software (AB Sciex, Singapore City, Singapore). The ESI source operation parameters were as follows: ion source, turbo spray; source temperature, 550 °C; ion spray voltage, 5500 V (positive ion mode)/−4500 V (negative ion mode); ion source, gas I and gas II; curtain gas were set at 50, 60, and 25.0 psi, respectively; the collision-activated dissociation was high. Instrument tuning and mass calibration were performed with 10 and 100 μmol/L polypropylene glycol solutions in QQQ and LIT modes, respectively. QQQ scans were acquired with collision gas (nitrogen) set to medium. A specific set of multiple reaction monitoring transitions were monitored for each period according to the metabolites eluted within this period. The relevant chromatograms are provided in Appendix A.

The identified metabolites were analyzed by PlantCyc (http://www.plantcyc.org/; accessed on 1 November 2020) and KEGG (http://www.kegg.jp/; accessed on 1 November 2020). OPLS-DA analysis was used to identify the differentially accumulated metabolites (DAMs), and the screening condition was |log2 (fold change)| ≥ 1. Each sample was performed in three biological replicates. Metabolome data have been submitted to additional file (Appendix A).

### 3.4. Transcriptome Sequencing

The raw RNA-seq data for walnut endopleura samples generated in our previous study are available with the bioproject accession PRJNA643637 (SRR15651947-SRR15651961) [38]. Transcriptome and metabolome samples were from the same batch of samples.

### 3.5. RNA-Seq Data Analysis

The walnut reference genome and gene model annotation files were downloaded from the website (http://www.xhhuanglab.cn/data/juglans.html; accessed on 1 November 2020) [39]. Clean reads were obtained by filtering raw reads using Perl scripts. Then, the clean reads were compared with the reference genome using Hisat2 software (v2.0.5, Johns Hopkins Unversity, Baltimore, MD, USA), and the number of reads mapped to each gene was counted by featurets v1.5.0-p3 to obtain the FPKM value. The differentially expressed genes (DEGs) were analyzed by OmicShare tool (www.omicshare.com/tools; accessed on 1 January 2021) (|log2 (fold change)| ≥ 1, FDR ≤ 0.05). KEGG enrichment analysis was performed using STRING database (http://www.string-db.org; accessed on 1 January 2021).

### 3.6. Co-Expression Network Analysis

Weighted gene co-expression network analysis (WGCNA) (V1.69) in R software package was used to construct the gene co-expression network, using the signed-hybrid network type. The co-expression network was mapped using Cytoscape V3.7.1 (https://cytoscape.org/ accessed on 1 January 2021) software. The description of gene function comes from the STRING database.

### 3.7. qRT-PCR Analysis

First, 0.5 μg RNA was reverse transcribed into single-stranded cDNA using PrimeScript RT Master Mix (Takara Biotechnology Co., Dalian, China). Real-time quantitative PCR experiments were then performed on ABI 7500 Real-Time PCR system (Applied Biosystems Inc., Foster City, CA, USA) with TB Green Premix Ex Taq (Takara). The instrument settings were: 95 °C for 30 s; 40 PCR cycles, with each cycle set at 94 °C for 5 s and 60 °C for 34 s. The specific primer information is shown in Appendix A, in which the *GAPDH* gene of walnut was used as the reference gene. The relative expression levels were calculated using the 2^−ΔΔCt^ method. Three biological replicates were performed.

## 4. Conclusions

In summary, a total of 760 metabolites were detected in the metabolome. The types and content of polyphenols in endopleura were more than those in embryos, and the content of polyphenols in endopleura was the highest at the mature stage. Phenylalanine metabolic pathway analysis showed that phenylalanine was gradually transformed into secondary metabolites during endopleura development. A total of 49 unigenes related to polyphenol synthesis were identified by endopleura transcriptome analysis. The expression patterns of *PAL*, *C4H*, *4CL*, *CHS*, *CHI*, *F3H*, *LDOX,* and *ANR* were similar, and the highest expression levels were at the mature stage. The results of transcriptome and metabolome were consistent, indicating that the mature stage was the key stage for polyphenol synthesis in endopleura. The transcription factor MYB111 played an important role in the synthesis of polyphenols, and its expression pattern was positively correlated with the accumulation pattern of quercetin, kaempferol, and procyanidins. Our study provides a comprehensive molecular biology background for the study of walnut endopleura development. It is helpful to study the synthesis mechanism of metabolites in walnut endopleura.

## Figures and Tables

**Figure 1 ijms-23-06623-f001:**
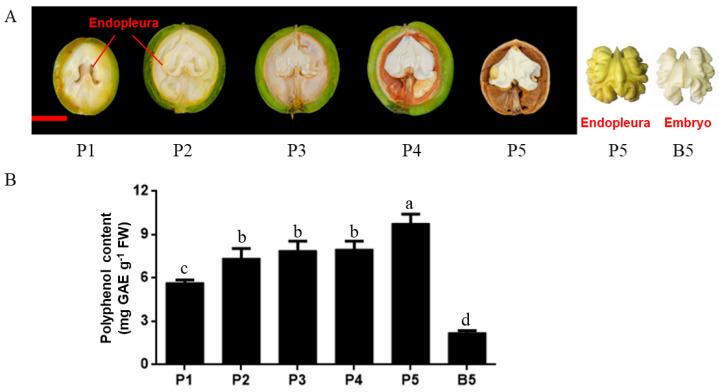
Morphological characteristics and polyphenol contents in the developing endopleura and mature embryo. (**A**) The image with black background shows the longitudinal section of the walnut fruits when the endopleura was sampled. Fruits were collected at 35, 63, 91, 119, and 147 DAP. The image with white background shows the endopleura and embryo at mature stage. The ruler is 2 cm. (**B**) Polyphenol contents in the developing endopleura and mature embryo. Values are means ± standard deviation (SD), n = 3. Same letters stand for insignificance at *p* ≥ 0.05 by one-way ANOVA.

**Figure 2 ijms-23-06623-f002:**
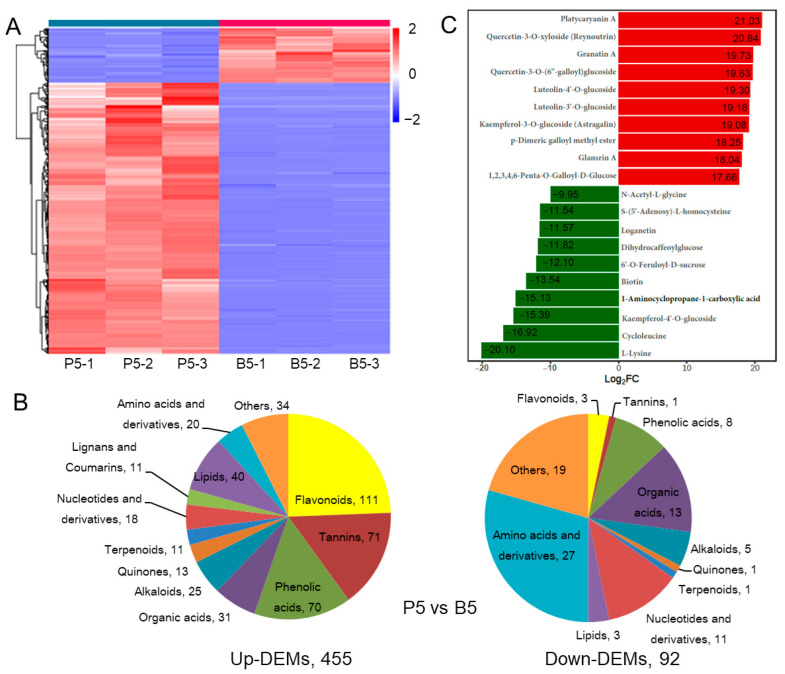
Differentially accumulated metabolites (DAMs) between endopleura and embryo in mature nut. (**A**) The heat map of DAMs in P5 vs. B5. (**B**) The classification of up- and down-DAMs, and the number of each classification. (**C**) Twenty metabolites with the greatest difference in content between endopleura and embryo.

**Figure 3 ijms-23-06623-f003:**
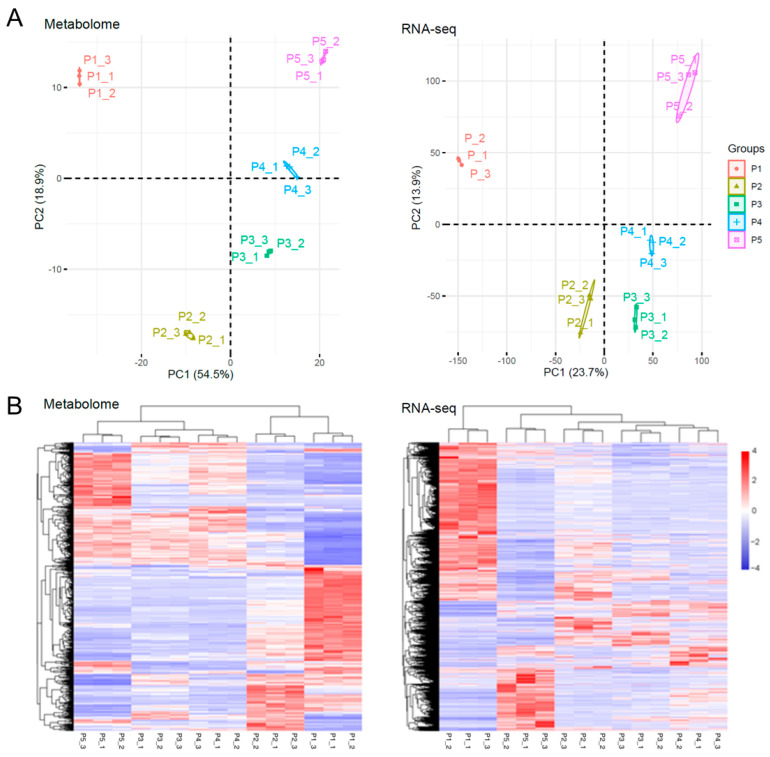
PCA of samples and cluster analysis of DAMs and DEGs in endopleura. (**A**) PCA of all 15 endopleura samples was conducted based on the relative contents of all detected metabolites (**left**) and RNAseq FPKM (**right**). (**B**) Cluster analysis of DAMs (**left**) and DEGs (**right**). The color scale (−4 to 4) represents the calculated Z-score.

**Figure 4 ijms-23-06623-f004:**
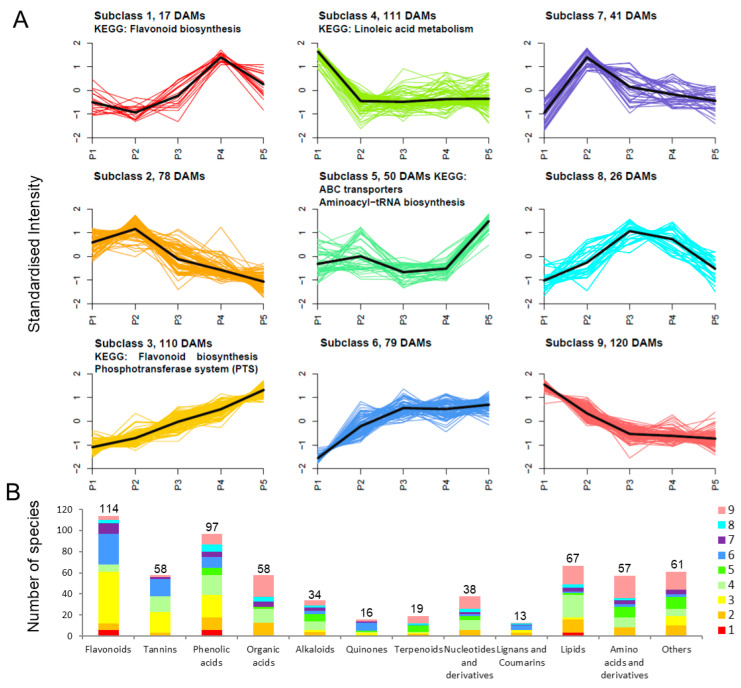
K-means diagram of differential metabolites and classification statistics. (**A**) Clustered DAM profiles in developing endopleura. The clusters were defined on the basis of metabolite content profile using the k-means method in R. (**B**) Quantity statistics based on metabolite species. The colors come from the classification of the subclass.

**Figure 5 ijms-23-06623-f005:**
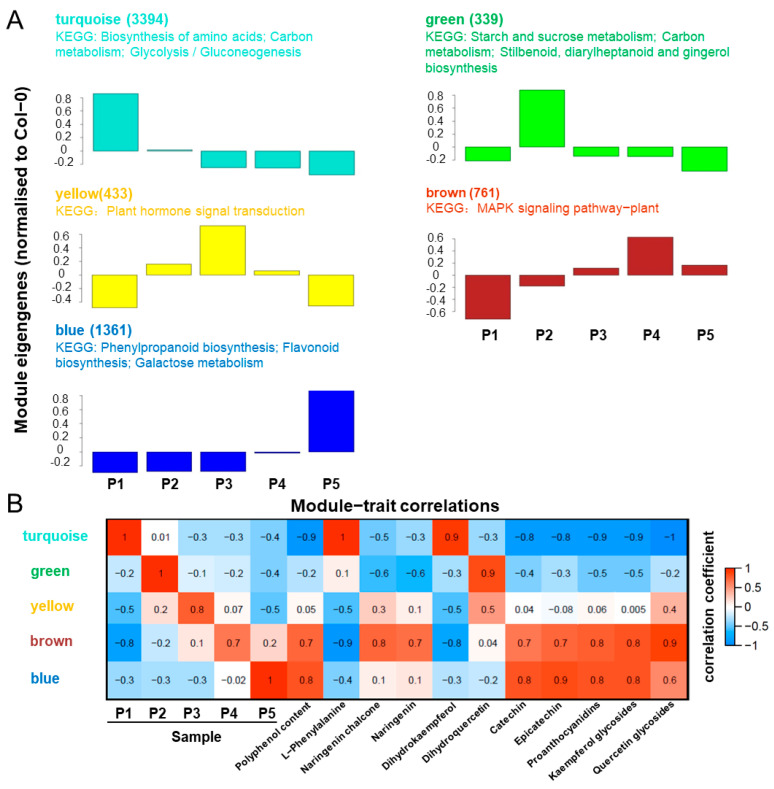
WGCNA of RNA-seq and metabolites. (**A**) WGCNA was calculated by 15 samples and the 6300 DEGs were classified into five modules. Columns represent module eigengene of mean values. The number of DEGs and KEGG pathways for each module are listed. (**B**) Expression patterns of the modules were correlated to samples and metabolites. The numbers in each colored box give the values for the correlation coefficient.

**Figure 6 ijms-23-06623-f006:**
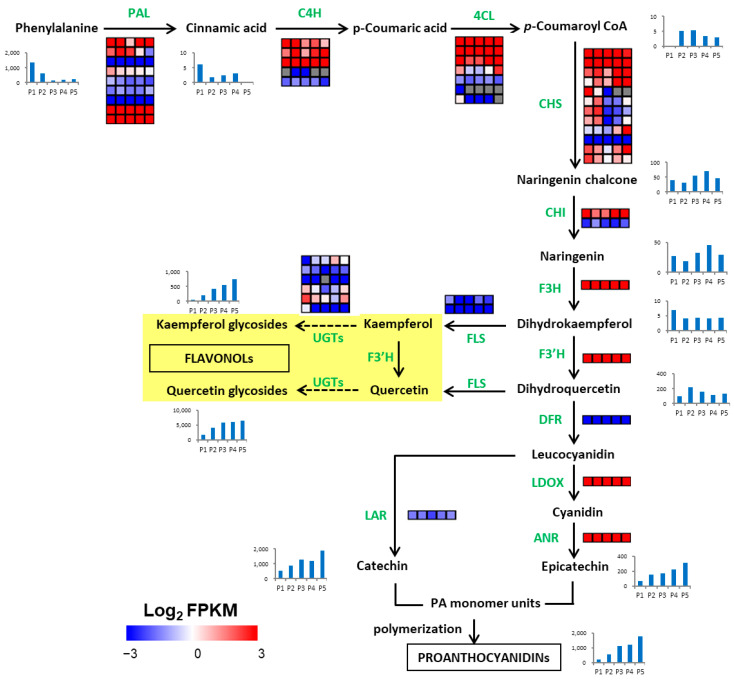
Transcriptional model of polyphenol biosynthesis in the developing walnut endopleura. The descriptions of genes is listed in Table 3. The rows represent different genes encoding the same enzyme, and the five squares in each horizontal row correspond to five stages (P1, P2, P3, P4, and P5). Grey squares mean FPKM = 0. The blue bar graph indicates the change in the content of metabolites in the metabolome (Table 2), and the *Y*-axis of the bar graph shows the relative content.

**Figure 7 ijms-23-06623-f007:**
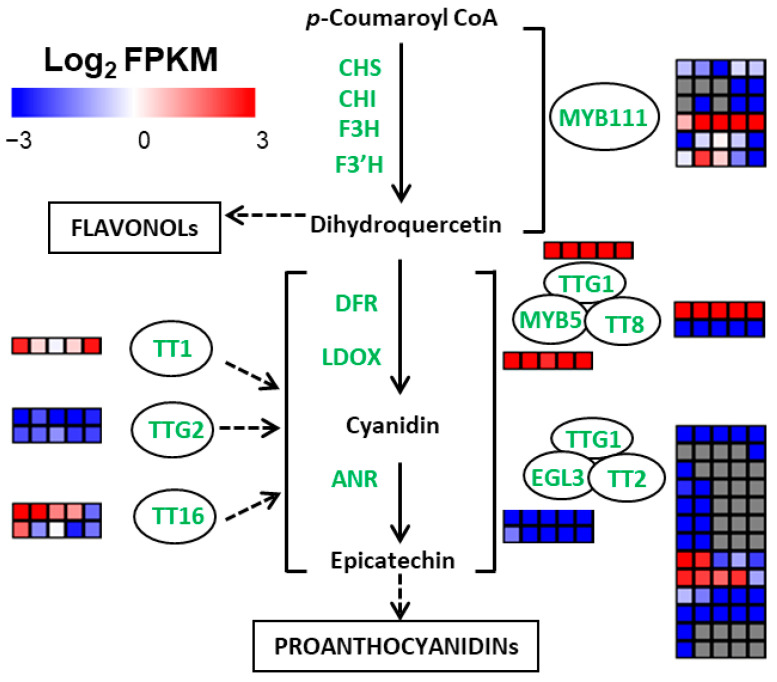
Regulation model for polyphenol biosynthesis by transcription factors in walnut endopleura. The descriptions of genes are listed in Table 4. The rows represent different genes encoding the same enzyme, and the five squares in each horizontal row correspond to five stages (P1, P2, P3, P4, and P5). Grey squares mean FPKM = 0.

**Figure 8 ijms-23-06623-f008:**
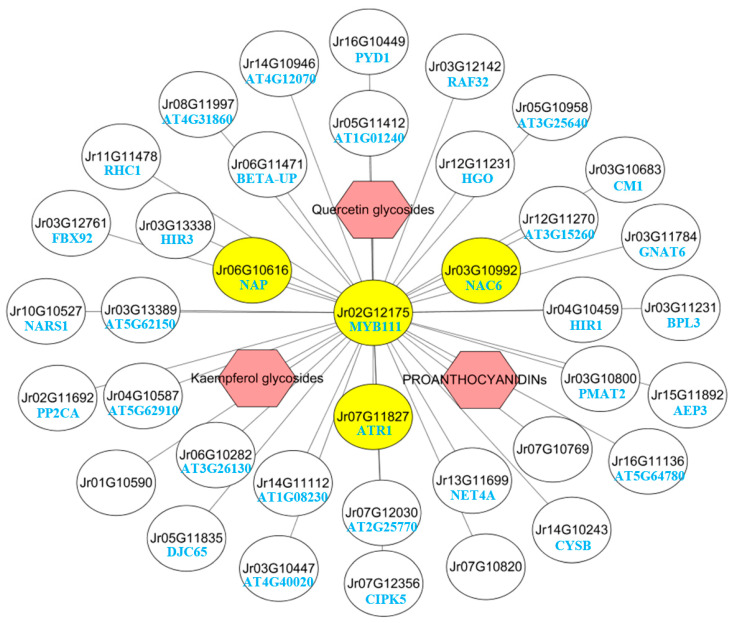
Co-expression network for *MYB111*(Jr02G12175). Using the WGCNA filtered gene list, we ranked the genes based on the weighted correlation weight with *MYB111* (Jr02G12175), and selected the top 37 genes from the blue module. The yellow solid circles represent transcription factors and the red hexagons represent major metabolites. The function of the genes is described in Appendix A.

**Figure 9 ijms-23-06623-f009:**
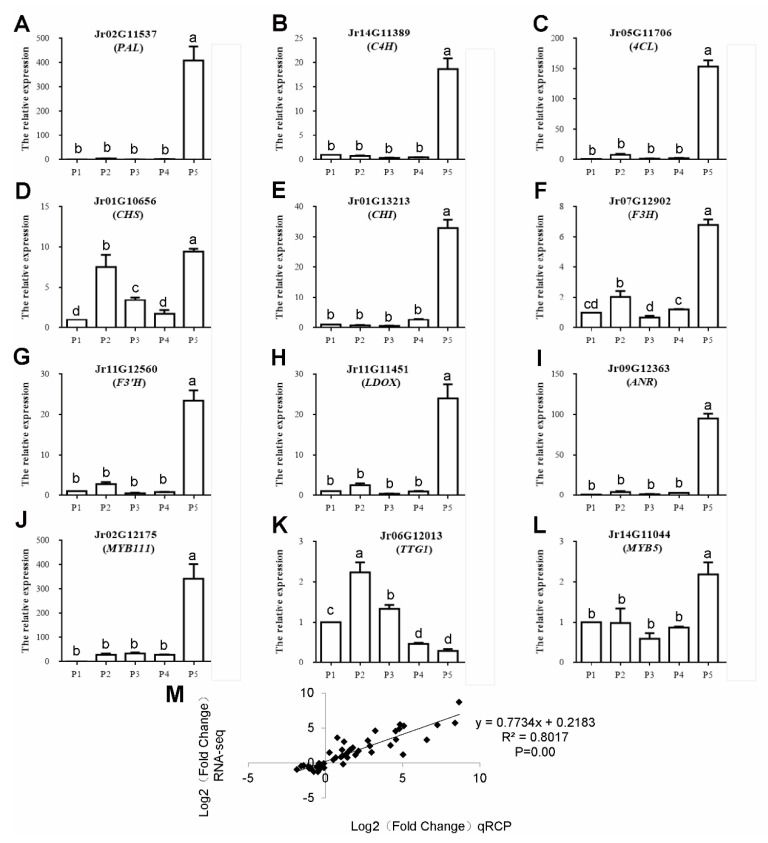
qPCR validation of differential expression. (**A**–**L**) qPCR of 12 important genes associated with polyphenol biosynthesis. Symbols represent mean values and short vertical lines indicate SE (n = 3). Same letters stand for statistically insignificance (*p* > 0.05). The unit of measure on the *x*-axis is sample. (**M**) A comparison of the gene expression ratios obtained from RNA-seq data and q-PCR.

**Table 1 ijms-23-06623-t001:** KEGG enrichment analysis of the DAMs between endopleura and embryo in the mature nut.

KEGG Pathway	Number of DAMs	*p*-Value
Flavonoid biosynthesis	28	0.0006
Flavone and flavonol biosynthesis	15	0.0338
Arginine and proline metabolism	14	0.0451

**Table 2 ijms-23-06623-t002:** Identification of polyphenol-related metabolites in the developing walnut endopleura. The compounds with values > 100 × 10^4^ are highlighted with bold font. The nine color subclasses are from Figure 4.

Compounds	Molecular Weight (Da)	P1	P2	P3	P4	P5	Subclass
		×10^4^	×10^4^	×10^4^	×10^4^	×10^4^	
Phenylalanine	165	**1367.73**	**613.52**	**152.69**	**186.15**	**230.17**	9
Naringenin chalcone	272	40.63	30.8	55.69	70.03	47.05	1
Naringenin (5,7,4′-trihydroxyflavanone)	272	27.55	19.02	32.15	45.58	29.86	1
Aromadendrin (dihydrokaempferol)	288	6.87	4.01	4.34	4.12	4.3	
Dihydroquercetin(taxifolin)	304	94.73	**221.76**	**156.95**	**116.33**	**130.31**	
Catechin	290	**533.24**	**875.19**	**1288.7**	**1184.77**	**1861.27**	3
Epicatechin	290	64.98	**153.18**	**167.48**	**220.96**	**311.58**	3
**PROANTHOCYANIDINs**							
Procyanidin B3	578	15.78	**117.61**	**290.39**	**269.8**	**400.24**	3
(EC→EC)g(1)	730	45.41	79.09	**147.32**	**170.2**	**187.31**	3
Procyanidin B2	578	61.04	**111.94**	**152.08**	**174.53**	**348.35**	3
Procyanidin B1	578	11.41	83.99	**186.75**	**175.01**	**276.04**	3
Procyanidin B4	578	10.81	21.49	68.08	91.78	**141.51**	3
Galloylprocyanidin B4	730	34.51	72.99	**143.77**	**137.29**	**204.11**	
Procyanidin C1 3′-O-gallate	1018	8.82	24.19	27.92	29.96	31.35	
Pedunculagin	784	1.83	8.78	32.76	62.05	64.32	3
Procyanidin C2	866	7.31	13.29	22.66	30.32	44.74	3
Procyanidin A6	592	5.33	5.56	13.58	13.34	25.18	3
Galloylprocyanidin C2	1018	0.66	3.53	4.41	5.47	5.95	6
Procyanidin C1	866	1.22	5.59	12.65	8.59	13.97	6
(EC→EC→EC)g(4)	1018	0.77	1.45	4.26	2.89	4.33	
(EC→EC→EC)g(5)	1018	0	3.12	4.4	3.02	4.09	6
(EC→EC→EC)g(3)	1018	0.79	2.12	3.11	5.09	3.54	
**Kaempferol glycosides**							
Kaempferol-4′-O-glucoside	448	11.68	0	0	0	0	
Kaempferol-3,7-O-dirhamnoside (kaempferitrin)	578	17.14	47.25	**102.78**	**120.58**	**203.56**	3
Kaempferol-3-O-(6″-acetyl)glucosyl-(1→3)-Galactoside	652	0	1.85	3.93	4.13	10.45	3
Kaempferol-3-O-rhamnoside (afzelin) (kaempferin)	432	6.81	10.54	8.72	10.5	5.71	
Kaempferol-3-O-glucoside (astragalin)	448	11.93	**134.91**	**284.47**	**385.36**	**497.94**	3
Kaempferol-3-O-(6″-p-coumaroyl)glucoside (Tiliroside)	594	0	1.78	2.57	1.84	2.84	6
Kaempferol-3-O-rutinoside (nicotiflorin)	594	0	1.1	4.16	11.91	13.58	3
Kaempferol-3,7-di-O-glucoside	610	0	1.78	6.89	8.68	9.77	3
Kaempferol-3-O-(6″-malonyl)glucoside	534	0	0	1.13	1.68	1.54	3
Kaempferol-3-O-sophoroside	610	0.66	0.97	1.04	0.82	0.79	
**Quercetin glycosides**							
Avicularin(quercetin-3-O-α-L-arabinofuranoside)	434	**188.08**	**754.09**	**1127.83**	**1422.17**	**1467.53**	6
Quercetin-3-O-(6″-malonyl)glucoside	550	0	0	0	1.69	1.47	1
Quercetin-4′-O-glucoside (spiraeoside)	464	26.09	56.66	**109.49**	**164.76**	**204.97**	3
Quercetin-7-O-glucoside	464	18.68	44.89	86.32	**117.17**	**141.43**	3
Quercetin-3-O-glucoside (isoquercitrin)	464	28.33	64.87	95.64	**127.38**	**103.48**	6
Quercetin-3-O-rhamnoside (quercitrin)	448	**607.33**	**1176.8**	**1408.7**	**880.52**	**984.4**	
Quercetin-3-O-galactoside (hyperin)	464	34.74	94.1	**135.54**	**162.06**	**138.69**	6
Quercetin-3-O-(6″-galloyl)galactoside	616	48.03	**346.3**	**630.69**	**649.71**	**805.03**	6
Quercetin-3-O-(2″,3″-digalloyl)-glucoside	768	**366.24**	**258.37**	**230.15**	**235.74**	**148.62**	9
Quercetin-7-O-rutinoside	610	6.92	1.83	4.83	5.15	6.42	4
Quercetin-3-O-xyloside (reynoutrin)	434	**223.32**	**865.72**	**1326.27**	**1676.7**	**1694.33**	6
Quercetin-3-O-(6″-galloyl)glucoside	616	45.85	**327.28**	**600.16**	**667.84**	**727.71**	6
Quercetin-3-O-(2″-O-rhamnosyl)galactoside	610	0	2.17	1.45	1.57	2.07	6
Quercetin-5-O-glucuronide	478	4.35	5.91	16.55	8.05	10.25	
Quercetin-3-O-sambubioside	596	1.28	0	0	4.41	6.23	3
Quercetin-3-O-(6″-p-coumaroyl)glucoside	610	0	2.25	2.91	1.99	4.16	6
Quercetin-7-O-(6″-malonyl)glucoside	550	1.21	4.2	4.03	3.04	2.21	7
Quercetin-3-O-glucosyl(1→4)rhamnoside-7-O-rutinoside	918	15.81	17.68	12.23	9.29	7.12	2
Quercetin-3-O-(6″-malonyl)glucosyl-5-O-glucoside	712	0	0.51	0.96	0.64	0.85	6
Quercetin-3,4′-O-di-glucoside	626	0	1.29	5.57	5.9	8.48	3
Quercetin-3-O-(2″-O-glucosyl)glucuronide	640	1.84	2.71	2.04	0	0	2

**Table 3 ijms-23-06623-t003:** Identification of genes involved in polyphenol synthesis. The expressed unigenes with FPKM values > 100 are highlighted with bold font. The three color modules are from Figure 5. “Jr” in the gene ID is an abbreviation of “JreChr”.

Enzyme	KEGG Annotation	Gene ID	P1	P2	P3	P4	P5	Model
PAL	Phenylalanine ammonia-lyase (EC:4.3.1.24)	Jr01G13110	11.28	15.65	1.63	15.62	50.56	Blue
		Jr04G10983	3.65	6.69	5	1.31	0.35	Turquoise
		Jr04G10985	0.02	0.04	0.04	0.03	0.04	
		Jr04G10989	2.25	1.45	1.3	1.19	0.91	
		Jr04G10993	0.55	0.39	0.28	0.32	0.23	
		Jr04G10994	0.32	0.45	0.34	0.34	0.52	
		Jr05G10885	0.08	0.06	0.03	0.03	0.02	
		Jr02G11537	18.64	62.06	21.79	68.69	**194.65**	Blue
		Jr09G11096	12.2	15.21	18.93	24.19	13.33	
C4H	Trans-cinnamate 4-monooxygenase (EC:1.14.14.91)	Jr13G11700	**103.25**	44.02	2.61	4.1	1.73	Turquoise
		Jr13G11701	3.12	7.9	2.1	5.84	15.49	Blue
		Jr14G11389	37.81	29.37	27.36	65.56	**213.3**	Blue
		Jr15G11420	0	0.05	0.01	0	0	
		Jr16G10525	0.28	0.39	0.32	0.35	0.18	
4CL	4-coumarate--CoA ligase (EC:6.2.1.12)	Jr10G10964	7.67	9.57	19.39	14.77	6.07	Yellow
		Jr13G10323	22.95	28.85	44.82	27	56.86	Blue
		Jr14G10240	3.71	5.53	2.07	8.09	7.01	
		Jr05G11706	1.11	3.23	1.89	9.06	47.92	Blue
		Jr05G12962	48.67	1.18	0.06	0	0	Turquoise
		Jr02G11900	1.2	3.48	0.31	0.33	1.11	
		Jr01G13626	3.46	5.47	0.18	0.24	0.47	Turquoise
		Jr02G11899	1.98	3.33	0.01	0.3	1.1	
		Jr07G11893	0.74	0.67	0.45	1.88	8.01	Blue
		Jr07G11894	0.12	0.01	0.01	0.03	0.09	
		Jr07G11896	4.26	2.44	0.78	2.9	9.96	Blue
		Jr11G10952	5.01	2.19	0.93	1.95	1.19	Turquoise
CHS	Chalcone synthase (EC:2.3.1.74)	Jr01G10656	39.87	**211.03**	**179.28**	**478.44**	**965.23**	Blue
		Jr02G10304	28.19	98.02	46.99	**187.99**	**330.66**	Blue
		Jr03G13232	10.32	4.12	4.15	13.14	8.52	
		Jr07G12835	2.25	0.64	0.55	1.11	5.33	Blue
		Jr15G11549	0.41	0.28	0.35	0.4	0.26	
		Jr16G11310	0.01	0	0	0	0	
		Jr16G11311	1.21	0.07	0.03	0.03	0	
CHI	Chalcone isomerase (EC:5.5.1.6)	Jr01G13213	7.66	3.23	3.29	12.99	17.29	Blue
		Jr02G11831	0.18	0.4	0.19	0.17	0.26	
F3H	Naringenin 3-dioxygenase (EC:1.14.11.9)	Jr07G12902	83.89	**151.38**	55.6	**235.15**	**755.52**	Blue
FLS	Flavonol synthase (EC:1.14.20.6)	Jr04G11647	0.4	0.03	0.04	0.3	0.2	
		Jr05G11778	0.13	0.03	0.01	0.01	0.02	
F3′H	Flavonoid 3′-monooxygenase (EC:1.14.14.82)	Jr11G12560	**191.32**	**204.87**	**122.11**	**109.43**	**199.86**	
DFR	Dihydroflavonol 4-reductase (EC:1.1.1.2341.1.1.219)	Jr07G11524	0.11	0.07	0.04	0.12	0.06	
LDOX	Anthocyanidin synthase (EC:1.14.20.4)	Jr11G11451	26.04	57.19	20.47	79.76	**263.77**	Blue
ANR	Anthocyanidin reductase (EC:1.3.1.77)	Jr09G12363	36.21	80.56	49.02	**132.8**	**358.59**	Blue
LAR	Leucoanthocyanidin reductase (EC:1.17.1.3)	Jr16G10851	0.34	0.34	0.22	0.32	0.39	
UGTs	Flavonol-3-O-L-rhamnoside-7-O-glucosyltransferase (EC:2.4.1.-)	Jr09G11788	0.05	0.63	0.69	1.81	1	
		Jr09G11865	0.42	0.29	0.13	0.22	0.28	
	Flavonol 3-O-glucosyltransferase (EC:2.4.1.91)	Jr04G11320	0.1	0.01	0	0.01	0.03	
		Jr04G11321	1.74	0.84	0.26	0.54	1.4	
		Jr04G11325	4.67	1.76	1.18	0.57	2.51	
		Jr08G11705	0.84	0.11	0.11	0.13	0.06	

**Table 4 ijms-23-06623-t004:** Identification of transcription factors involved in polyphenol synthesis. The four color modules are from Figure 5.

Enzyme	KEGG Annotation	Gene ID	P1	P2	P3	P4	P5	Model
MYB111	myb domain protein 111	Jr01G10083	0.57	0.38	0.1	0.72	0.62	
		Jr02G10456	0	0	0	0.02	0	
		Jr02G12175	1.89	56.08	73.12	83.57	97.4	Brown
		Jr07G11017	0.01	0.7	1.11	0.64	0.01	
		Jr08G10063	0.83	4.93	1.53	0.35	0.01	Green
TTG1	Transducin/WD40 repeat-like	Jr06G12013	74.18	65.96	71.07	52.08	38.58	
TT8	Basic helix-loop-helix (bHLH)	Jr07G10705	15.61	10.83	13.3	14.57	11.47	
		Jr08G10317	0.03	0.01	0.01	0.01	0.01	
TT2	Duplicated homeodomain-like superfamily	Jr01G12462	0	0	0.01	0.12	0	
		Jr02G11138	0	0	0	0	0.01	
		Jr03G11815	0.18	0.02	0	0	0	
		Jr03G11816	0	0.03	0	0	0	
		Jr03G11818	0.05	0.01	0	0	0	
		Jr03G11824	0.07	0.01	0	0	0	
		Jr03G11832	8.17	5.69	0.23	0.47	0.21	Turquoise
		Jr03G11834	5.95	4.7	3.57	5.23	0.47	Turquoise
		Jr04G10961	0.55	0.34	0.12	0.09	0.06	
		Jr08G11467	0.01	0	0	0	0	
		Jr08G11642	0.01	0	0	0	0	
MYB5	myb domain protein 5	Jr14G11044	12.12	7.57	5.49	10.61	24.69	Blue
EGL3	Basic helix-loop-helix (bHLH)	Jr11G11341	0.01	0.01	0	0.01	0.01	
		Jr12G10985	0.33	0.11	0.09	0.08	0.12	
TT1	C2H2 and C2HC zinc fingers	Jr08G11835	5.7	1.42	0.91	1.47	6.41	Blue
TT16	K-box region and MADS-box	Jr05G10903	42.69	8.38	2.85	2.4	0.3	Turquoise
		Jr06G10192	3.56	0.39	0.94	0.17	0.31	
TTG2	WRKY family	Jr09G11240	0.09	0.24	0.09	0.08	0.17	
		Jr10G11149	0.23	0.28	0.4	0.2	0.22	

## Data Availability

Not applicable.

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
