# Peer review of "Metabolome and Transcriptome Profiling Unveil the Mechanisms of Polyphenol Synthesis in the Developing Endopleura of Walnut (Juglans regia L.)"

_ijms, 2022, doi:10.3390/ijms23126623_

Round 1

Reviewer 1 Report

Dear authors, thank you for the excellent work! The article presents new and useful information. I only recommend a few things:

Lines 12 and 13: “Walnut (Juglans regia L.) is an important woody nut tree species, and its endopleura 12 (the inner coating of a seed) is rich in many polyphenols.”

Line 31: “and insights regarding the”

Line 39: “nutrients, such as phenolic acids and”

Line 46: “a strong”

Line 47: “can enhance immunity, provide resistance against atherosclerosis and protect eyesight”

Lines 89 and 90: change to “accumulate”, no?

Numerate sections and subsections, please.

Line 119: Why did you decide to use acetone extracts in the Folin-Ciocalteu method, but methanol in the other assays? This is not correct, I recommend to perform again the Folin-Ciocalteu method using methanol, please.

All the assays were done, at least, in triplicate, correct? Mention this fact in section 2 and not in the discussion.

Line 204: “concentration in the embryo was about 7 times ???? that in endopleura” higher or lower?

Line 265: “naringin”

Is it correct to claim that the accumulation of walnuts is intimately correlated to the biological potential of walnuts? I think that making, at least, a DPPH assay, for example, will be an added value to understanding it.

Please verify the letter type throughout the article.

I think that will be an added value to insert a table with the quantity of each identified phenolic compound, in order to make the article clearer.

Uniformize endopleura (or Endopleura) throughout the article.

Reviewer 2 Report

The subject of the manuscript is very interesting.  Study provided basic data for the research of the metabolite species and the  formation mechanism of walnut endopleura. It can be considered for publication after some minor corrections.

General Comment:

According to authors all analysis methods  (UPLC ,spectrophotometric)are already available in literature sources. Never the less this work is missing information how all methods were tested to ensure proper results are obtained. At least basic method transfer is needed to check specificity, range, repeatability etc. I believe such tests were performed but not described as method development was not subject of this work. Please include information how literature methods performance in this specific use were evaluated. This is especially important for HPLC and spectrophotometric methods. A summary of these activities should be included.

Specific Comments:

Line 98 (Materials and Methods): Please indicate the reagents and chemicals as well as individual standards for the polyphenols being determined and their source of origin.

Line 124- Please provide standard curve for individual spectrophotometric method

Line 135- Please provide a brief description of the UPLC methods. Please indicate which mobile phase was used, flow and wavelength. No validation of the HPLC method (selectivity, precision, repeatability, etc.)
